# Commercial Level Analysis of P2P vs. Net-Metering Comparing Economic and Technical Indexes

Esteban A. Soto [1] , Alexander Vizcarrondo Ortega [2], Andrea Hernandez [2] and Lisa Bosman [3,*]

1 Department of Engineering and Technology, Southeast Missouri State University, Cape Girardeau, MO 63701, USA; esotovera@semo.edu
2 Department of Industrial Engineering, University of Puerto Rico at Mayaguez, Mayaguez, PR 00680, USA; alexander.vizcarrondo@upr.edu (A.V.O.); andrea.hernandez19@upr.edu (A.H.)
3 Purdue Polytechnic Institute, Purdue University, West Lafayette, IN 47907, USA
* Correspondence: lbosman@purdue.edu

**Abstract:** As photovoltaics (PV), also known as solar electricity, has been growing over the years, the energy markets have been gradually moving toward decentralization. However, recent media accusations suggest that decentralized renewable energy is slowly becoming unpopular because of the hidden fees being charged to owners of installed PV systems. In response, this paper investigates the potential for alternative approaches to incentivize owners using peer-to-peer (P2P) sharing. This study provides an analytical comparison between the use of the P2P mechanism, the net-metering mechanism, and a combination of these in the commercial sector. Through the use of a simulation, this case study presents the possible outcomes of the implementation of these models in a microgrid. Using technical and economic indexes the comparison was made by looking at the following indexes: peak power, energy balance, economic benefit, and transaction index. Based on a microgrid of 28 commercial buildings, readings of consumption were taken at intervals of one hour, and a Python model was made to find PV size and compare trading mechanisms. It was found that the combination of P2P and net-metering had the best overall performance, followed by net-metering itself, with the best season being all for both, and summer for net-metering by itself. This shows that a P2P model implemented in a microgrid helps create more energy balance, although the combination would achieve the highest performance. This study can be used by policymakers for proposing renewable energy policies and regulations that are more beneficial to all prosumers and consumers.

**Keywords:** solar energy; PV systems; microgrids; performance





## 1. Introduction

### 1.1. Problem Identification

There has been undeniable growth in the energy market in recent years, which has increased distributed energy generation. As a result, the industry is gradually moving toward a more decentralized energy market. The photovoltaic (PV) market share has steadily increased since 2013, with solar accounting for 46% of all new electricity capacity added to the grid in the United States in 2021 [1]. The adoption of solar generation by large corporations has accelerated the growth of solar energy even further. However, the commercial solar market accounts for slightly more than 1% of total commercial energy demand, implying that there is significant room for growth within this sector [1]. New energy models are required to incentivize and support the transition to clean energy by promoting energy market decentralization.

Furthermore, net-metering, one of the current options for an open energy market, is being phased out [2]. Net metering allows prosumers to inject the excess energy they generate back into the grid in exchange for compensation from the utility provider. Over 60 countries have implemented net metering at a structural level as of 2018 based on the REN21 global status report [3]. Nevertheless, net metering programs worldwide have

seen some rollback in the past few years. Some countries have started charging prosumers fees for participating in their net metering programs. For instance, Egypt has declared it will start charging a "merger fee" to PV owners who take advantage of net-metering. Another example is the fee the Belgian region of Wallonia will start imposing on net-metering systems [4]. Some states in the US, such as Kentucky, New York, and Utah, have also modified their net metering policies to reduce their compensation for prosumers, including proposing alternatives for net metering and implementing additional fees [4]. It is anticipated that incentives for renewable energy will decline if this trend persists across nations. Furthermore, if no changes are made, current and future PV owners will experience a significant slowing in their return on investment, which will have an impact on the energy market.

### 1.2. Proposed Contribution to Literature

At least three studies [2,5,6] have presented peer-to-peer (P2P) as a beneficial mechanism for managing surpluses of energy in microgrids. The findings of Soto et al. indicate that the implementation of P2P can benefit prosumers economically [2]. On the other hand, Zhang et al. found that the integration of a P2P mechanism into a microgrid helps maintain a better energy balance in the grid [5]. The global energy costs for a set of peers were also reduced with the implementation of P2P in the study of Baez et al. [6]. Distributed energy resources have increased significantly in the past few years, which allows for the energy market to be decentralized [5], thus opening the energy market to alternatives such as P2P, where prosumers can sell their excess energy to peers in the community. Since the energy market will have multiple generators of electricity, a space for competitive electricity rates will be opened. P2P also helps maintain a better energy balance in the power grid [7]. In addition, since prosumers will have an additional income directly related to their PV system, their period of return on investment will accelerate. Considering all the gaps within the current alternatives for energy management, including the phasing out of net-metering, P2P can be a viable option for prosumers to benefit from their surpluses of electricity.

This paper proposes an analytical comparison between the use of the P2P mechanism, the net-metering mechanism, and a combination of these in the commercial sector. Through the use of a simulation, this case study will present the possible outcomes of the implementation of these models in a microgrid. In addition, this study presents the sensitivity of these mechanisms to the different seasons of the year. With the use of technical and economic indexes, P2P will be evaluated as a complement or an alternative to current net-metering policies. Moreover, these same indexes will provide evidence of how this model can benefit its users economically and technically. This study will encourage policymakers to create renewable energy policies and regulations that are more beneficial to all prosumers and consumers.

This study makes a substantial contribution to the field of decentralized energy markets and solar energy integration by offering a comprehensive comparative analysis of peer-to-peer (P2P) trading mechanisms and net-metering within the context of a microgrid environment focused on commercial buildings. The novelty of this paper extends beyond the mere application of an existing tool and the definition of metrics. The primary contribution lies in the holistic assessment and juxtaposition of multiple P2P trading mechanisms and net-metering, rather than a singular focus. The paper not only presents an innovative approach by combining these mechanisms but also delves into a meticulous examination of various technical and economic indexes, including peak power, energy balance, economic benefit, and transaction index. This methodological rigor ensures a robust comparison of the trading methods.

Furthermore, the contribution of the study is magnified through its engagement with the detailed data set of the consumption of 28 different commercial buildings and the potential generation of solar generation on-site. The investigation extends beyond tool utilization by considering practical implications and operational complexities, thereby elevating the relevance and applicability of the findings. The research's contextualization

within specific seasons and its identification of optimal performance conditions for both P2P and net-metering configurations provide actionable insights for energy market stakeholders and policymakers.

## 2. Literature Review

### 2.1. Net-Metering

Electrical consumers who also generate solar energy often participate in a billing arrangement, known as solar net-metering [8]. This arrangement allows the owner of the PV system to be compensated by the utility company at a specified rate for all the excess energy generated onsite by injecting it back into the grid. When the prosumer needs to draw energy from the grid, this compensation can be exchanged for energy from the utility company. In some cases, the rate is constant for consumption and generation, while in other cases, the rate for generation might triple the consumption rate [8] or be lower than the retail compensation rate. This compensation stands currently as one of the more significant incentives for the acquisition of PV systems. Hence, the compensation policies established by each state will directly impact consumers' inclination to adopt PV systems. Even though some utilities have chosen to freely offer net metering to their consumers, net metering is typically limited by state rules and regulations [9]. In addition to the traditional net metering policy, aggregate net-metering allows nonprofit organizations, municipalities, and multiple properties to benefit from the compensation offered for the excess energy generated [2]. Aggregate net-metering can be segmented into four different categories: basic meter aggregation, tenant aggregation, multiple site aggregation, and virtual net metering [9]. For multiple buildings owned by a single customer with separate meters but near one another, basic meter aggregation is possible. Another option is tenant aggregation if multiple customers have their meters on the same contiguous property. A multi-site aggregation is an option for single customers who have meters in geographically disconnected areas. Finally, virtual net metering (VNM) is the most versatile option for aggregate metering. VNM allows multiple customers with individual meters to share a common PV system and distribute the energy generated and energy compensations among themselves [2,9]. The advantage of this last option is that that energy distributed can be controlled and this allows consumers who cannot install a PV system by themselves to still benefit from it.

Globally, aggregate net-metering support programs have grown in recent years (see Figure 1). Net-metering policies have existed at the state level in the United States since the 1980s [10]. Over the years, different energy policies have been implemented. After Congress took action in 2005, 45 states have net metering policies in some form. A total of 28 States exercise aggregate metering, including California, Colorado, and Connecticut [8,9]. Based on recent studies, shared net-metering is more scalable and efficient than traditional net metering [11]. Net-metering programs are not only implemented in the US but in many locations across the world, such as Europe, Canada, Australia, India, and Pakistan. Denmark started its net metering program in 1998 and have currently a 1-h net metering program [12,13]. Other counties have incorporated other rules, such as feed-in tariffs and net billing with net-metering, including Portugal and Italy. In Greece, authors believe that the growth in the PV sector is mainly due to the net metering policies [14]. A constant pattern, independent of the country, is the direct relationship between the growth of the PV sector and the various supporting programs/strategies implemented.

Net metering is critical to the growth of the PV sector; however, it is not without its challenges. Studies have shown that a compensation rate at the retail price, which is often paid, is typically not optimal. It has been discovered that there is substantial variability between the optimal unit of payment for solar-distributed generation of electricity depending on the energy industry. It can be either more or less than the retail price [15]. Since states use different compensation rates or compensation schemes for the energy injected back into the grid by net-metering users [10], some environments are more hospitable for net-metering participants. Among the compensation schemes, some allow buying excess

energy for a rate below the retail rate, which might be discouraging for individuals who plan to acquire their own PV system. In addition, the main issue with net metering is the reduction of its benefits and accessibility. Most net-metering policies limit the amount of energy exchanged, therefore reducing the possible compensation [12]. Net metering users may get charged higher prices to subsidize the credits provided to net metering users, creating a backlash from utility-billed customers. Finally, it has been found that conventional net metering is incapable of serving all customer categories [11]. In other words, net-metering policies should be more flexible to encourage and benefit PV owners according to their particular situation, such as pattern of consumption or location.

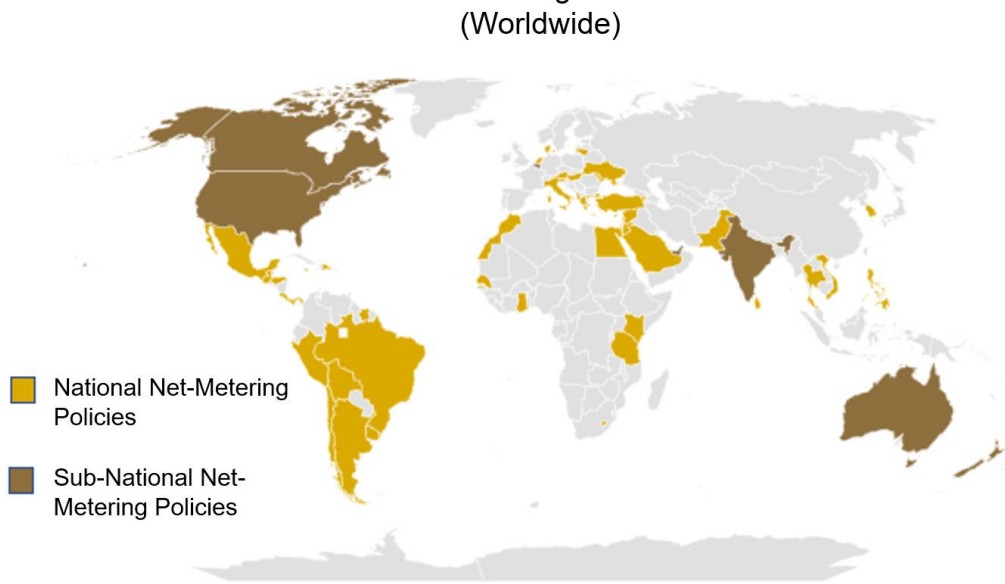

**Figure 1.** World map for existing net-metering policies.

### 2.2. P2P Commercial Area

The energy industry is constantly transforming, since new energy technologies and information technologies are being developed every day. In the past few years, the inclination of the energy industry toward a more decentralized market has been observed. This can be attributed to the deployment and increasing connection of distributed energy resources (DERs) from renewables [5]. This leads to a variation in traditional consumers where these become prosumers, consumers who both consume and generate energy. Prosumers have various options on how to allocate their excess energy generated. These include battery storage, net-metering programs, or selling their surplus of energy to other consumers. Peer-to-Peer energy trading is composed of a community of consumers and prosumers who exchange energy directly among themselves without the need for an intermediary [5,16]. The P2P model has several advantages. First, it helps reduce the energy bill of commercial buildings, as found by Zhang et al. in his study [17]. Second, this mechanism allows the setting of preferences and more transparent transactions, thus gaining the trust of its consumers. Third, when considering aggregated groups of prosumers who are physically dispersed, the variability of the electric load is significantly reduced when compared to a single type of generation [6]. Fourth, this idea benefits the community in which it operates by creating jobs and funding community assets [16]. Finally, having DER will relieve the pressure on the power grids and facilitate a better balance of energy [18].

P2P projects for energy exchange are being carried out across the world. Some are focused on business models and platforms while others target local control over micro-grids [19]. Countries such as Chile, France, Germany, and Italy have similar projects to P2P, but adapted to their energy market structure [4]. For instance, prosumers in Germany are legally allowed to sell their energy directly or through aggregators, although

if they wish to directly participate as a supplier a permit is required. In addition, they have multiple encouraging policies for collective prosumers, such as sharing electricity in the same multi-apartment building or neighborhood [20]. In 2020, Chile introduced the option of providing energy generated by small-scale PV systems to other consumers in the community [4]. Other countries, such as the US, Australia, and UK, have regulations that do not allow the implementation of P2P energy trading [16]. Nonetheless, many P2P pilot projects and trials have been tested in these same countries. For example, Yeloha and Mosai were trials developed in the US in which non-solar energy owners were allowed to pay for a portion of the solar energy generated by the host's solar system and obtain a reduction on their utility bills. Piclo was a partnership between an innovative technology company and renewable energy providers that allowed their consumers to buy electricity directly from local renewables [19].

The implementation of a P2P model will include some challenges. The biggest challenge will be the integration of regulations that allow and support P2P energy trading. In most countries, no regulations or policies have been established for the legal implementation of a P2P mechanism for energy trading [21]. For the government to adopt these regulations, they need to foresee P2P energy trading as an achievable and effective model to support renewable energies, an additional issue for the integration of the model as part of the public grid infrastructure and its role in the current energy market structure [16]. After this, a significant challenge is the development of a software platform that enables the safe and effective exchange of information, energy, and money among peers [2,5,21]. Defining the strategy for trading prices at the commercial level as the energy market becomes more decentralized will require additional research effort [22]. Most of the important considerations are related to data management and protection, technical challenges such as infrastructure and smart meters, and policy regulations [16,23].

## 3. Methods

### 3.1. Simulation Models

A microgrid simulation was created using two models. The microgrid considers only commercial buildings. Using technical and economic indexes, the following two models were compared to one another:

Net-metering: In this model (Figure 2), an arrangement is made between the electric company and the prosumers and customers under which energy is provided to them by the electric company if needed. An additional component of this arrangement is the option for the prosumer to sell excess energy generated by their photovoltaics back to the utility company for an agreed rate.

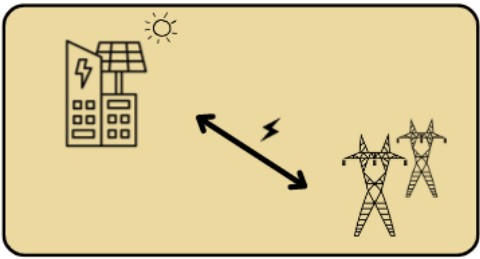

**Figure 2.** Net-Metering (Summarized).

P2P Model: An additional energy path is created in this model (Figure 3). The local grid will be used to facilitate energy exchange amongst peers because of this. In other words, there will be a flow of cash between prosumers and consumers in the local grid. Since prosumers are seen as owners of the microgrid, it is important to clarify that the expenses of using the grid are not taken into account.

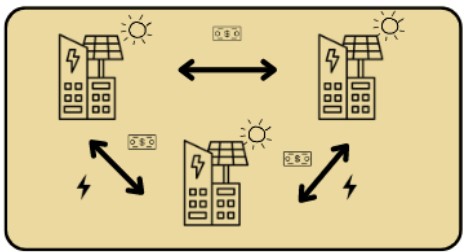

**Figure 3.** P2P (Summarized).

P2P combined with Net-metering: This model (Figure 4) includes an arrangement between prosumers and the electric company and energy trading among prosumers in a community. This path allows prosumers to sell their excess energy generated onsite to other peers in the community. In addition, prosumers will also have the option of selling any surpluses of energy not traded with peers, back to the electricity company. It should be noted that the expenses of using the grid for energy trading in the community are not considered.

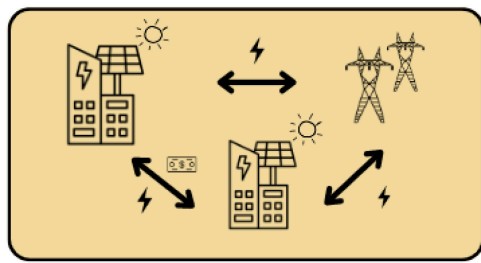

**Figure 4.** P2P Combined with Net-Metering (Summarized).

*3.2. Indexes*

To assess and contrast the two models, the usage of economic and technical indexes will be considered.

Peak power index: For each season, the peak of power was estimated over a specific time period. This metric is significant since it indicates that the system is under a heavy load at that time, which could result in significant energy imbalances. The peak of energy might be either negative or positive because of taking both energy surpluses and deficits into account. To make these comparable, the absolute value of both was used to determine the peak during that time period. The peak of energy is given by Equation (1):

$$\text{Peak of Power} = \text{Max}_T \left\{ \left| \sum \text{surplus}_{i,t} \right|, \quad \left| \sum \text{deficit}_{i,t} \right| \right\} \tag{1}$$

where the sub-indexes represent the following:

$$T = \text{Season}$$

$$I = \text{Day of Season}$$

$$t = \text{Time of Day}$$

The peak of power for each season was divided by the maximum value obtained from the peaks from all four seasons to standardize the index. It was necessary to calculate the inverse to determine our index's range, which is between 1 and 0. In this instance, a score

of 1 represented the best performance and a score of 0 the worst performance. The peak of power for each season is calculated using the following formula:

$$\text{Peak of Power Index} = 1 - \frac{\text{Peak of Power}}{\text{Max}_T[\text{Peak of Power}]} \tag{2}$$

Energy balance index: The purpose of this metric is to measure the amount of energy imported and exported into the grid with the bulk transmission system. The importance of this index lies in maximizing the use of all available energy. This microgrid is composed of 28 buildings connected to the bulk of power. To evaluate this index, consideration was placed on the surpluses of energy, the deficits of energy, the consumption and generation of energy, and, for P2P, the interchange of energy. An important clarification is that the term energy exchange only applies to P2P models because in net metering the interchange is always zero. On the other hand, in P2P models, when energy is sold and/or bought among the users, the interchange of energy happens. It should be noted that, given that the amount of energy bought in the microgrid is the same, each value is only considered one time. There are two ways in which energy imbalances can occur. First, when the solar energy generated surpasses the amount of energy consumed in the system, surpluses occur. When this happens, the excess energy generated is fed back into the microgrid system. Second, energy deficits, occur due to a greater amount of energy being consumed than the amount of solar energy generated in the microgrid. In other words, it happens due to the difference in an energy deficit and energy interchange in the microgrid. The numerator of the formula represents the energy imbalance by considering the difference between the maximum value between the surplus and deficit and the interchange carried out at a certain time (season). The denominator is the summation of consumption and the absolute value of generation, since in this case the generation value was considered negative. Equation (3) shows the formula for the calculation of Energy Imbalance:

$$\text{Energy Imbalance} = \frac{\sum_T \left[ \text{Max}_t \left\{ \left| \sum \text{surplus}_{i,t} \right|, \left| \sum \text{deficit}_{i,t} \right| \right\} - \sum \text{interchange}_{i,t} \right]}{\sum_T \left[ \sum \text{consumption}_{i,t} + \left| \sum \text{generation}_{i,t} \right| \right]} \tag{3}$$

Then, the Energy Balance index is determined using the formula below:

$$\text{Energy Balance} = 1 - \text{Energy Imbalance} \tag{4}$$

Economic Benefit: The general economic benefits index calculates the financial gains from utilizing a P2P mechanism. For modeling purposes, it is assumed that there will be a difference between the retail price—the price for the energy exchanged and consumed from the electrical network—and the export price—the price of excess energy exported to the electrical grid in the examined P2P model. The following equation is used to construct the economic benefit index:

$$\text{Economic Benefit} = \frac{\text{Value\_mechanism}_T - \text{Value\_ref}_T}{\text{Value\_max}_T - \text{Value\_ref}_T} \tag{5}$$

For the calculation of this index, the values considered were the economic value of the sale utilizing the model and the economic value obtained from net metering if used. The value of the mechanism (Value\_mechanism$_T$) corresponds to the possible economic value obtained from the sale of energy if utilizing the P2P model. On the other hand, the reference value (Value\_ref$_T$) is the minimal economic value obtained from the purchase's energy through the P2P model. Lastly, the maximum value (Value\_max$_T$) represents the maximum economic value when considering the highest price of energy possible in both models, P2P and net metering, in a determined time (season). In other words, this value of

each was calculated for each of the four seasons. It should be mentioned that for this model, the economic benefit was based on the entire system (microgrid) and not on each prosumer alone.

Transaction index: This index will be evaluated as two separate indexes. The first one, Transaction vs. Supply, measures the energy exchanged in the microgrid versus the total amount of energy offered considering the P2P market. The second, Transaction vs. Demand, measures the total amount of energy exchanged in the microgrid versus the total amount of energy needed within the P2P market.

$$\text{Transaction vs. Supply} = \frac{\sum \text{transactions}_{i,t}}{\sum \text{supply}_{i,t}} \tag{6}$$

$$\text{Transaction vs. Demand} = \frac{\sum \text{transactions}_{i,t}}{\sum \text{demand}_{i,t}} \tag{7}$$

In the above equations, the transactions show the overall energy traded in kWh over a given period. The demand is the amount of energy needed in kWh over a specific period. The supply lists the energy surpluses that are available for trading in the market in kWh or the energy surpluses.

### 3.3. Study Design

Two models were studied through the implementation of simulations. This case study was based on a microgrid composed of 28 representative commercial buildings situated in the states of Illinois and Indiana. The consumption profiles were created and extracted from the NREL 2018 database on Open Energy Data Initiative's website [24]. Additionally, the PVWatts calculator tool [25] was used to build the photovoltaic energy generation profiles. In both models, the data was evaluated by the hour, and the average cost of electricity for the commercial sector was based on data from Energy Information Administration [26]. Seven representative days, starting on Monday and ending on Sunday, from each season of the year—winter (January), spring (April), summer (July), and fall (October)—were simulated in the models per hour.

A total of 28 consumption profiles were created from the database of NREL from the year 2018 for the states of Illinois and Indiana. For this study, the readings of consumption were in intervals of one hour and were transformed to kilowatts per hour. On another hand, the NREL PVWatts tool was utilized to create the solar generation profiles. The locations for the solar energy profiles were chosen at random from the states of Illinois and Indiana. A total of 28 zip codes from both states were randomly selected. To determine the size of the PV systems, all the consumption readings for the year were added as shown in Equation (8):

$$\text{Total Consumption} = \sum_{T} \text{consumption}_{i,t} \tag{8}$$

After this, the average consumption of energy daily was calculated and transformed into kilowatts per hour with the following equation:

$$\text{Average Daily Consumption(kWh)} = \frac{\text{Total Consumption}}{365} \times \frac{1}{1000} \tag{9}$$

With his number, the power output was calculated using Equation (10):

$$\text{Power Output} = \frac{\text{Average Daily Consumption}}{\text{Annual Solar Radiation}} \tag{10}$$

Then, the power output was divided by the derate factor, and the PV system size was obtained using Equation (11):

$$\text{PV System Size} = \frac{\text{Power Output}}{0.86} \tag{11}$$

This study considers the average price of electricity to be 0.114 USD per kWh, which is the average price between the states of Indiana and Illinois. However, this study uses the price of 0.0982 USD per kWh for the net metering model, which is the lowest price for electricity between the two states mentioned above. In addition, the price for the P2P model varies between 0.0982 and 0.114 USD per kWh.

*3.4. Data Analysis*

A Python model was used for the simulation, with PyMarket as the main library. PyMarket is a Python library designed to simulate and compare various trading mechanisms, including peer-to-peer trading [27]. The purpose of this library is entirely for academic research and to establish a baseline for future multi-agent systems. One of the advantages of this library is the flexibility it allows its users according to the needs of their projects. It also features a sizable community of programmers who collaborate and work together to produce new and improved programs. However, due to the short amount of time, the library has been open source, and enhancements are made more slowly [2]. In the P2P models, the PyMarket library was used to generate the pairs and transactions of prosumers and consumers.

The PyMarket library's P2P model is based on two studies [28,29] in which a market is considered with two entities, buyers and sellers. Trades are conducted by pairing random participants. In this market, once bidding starts, no new entrants are allowed. To avoid competitive advantages, the trading order is random. In addition, because there is a limited amount of electricity available to trade, each transaction has different prices in this market. The PyMarket library has been used in P2P projects for local energy markets, to encourage the use of local renewable energy [30], which makes it most adequate for this simulation.

In this study, biddings are generated every hour and depend on the amount of energy consumed and generated. The energy offered in the bidding process is all the surpluses of energy and the energy demand is composed of all the energy deficits. In the P2P model, prosumers offer energy at random prices ranging from 0.0982 USD per kWh to 0.114 USD per kWh. The same prices are established for the energy demanded by consumers. The way the bidding works is that, if the amount the consumer bids exceeds the amount asked by the prosumer, the transaction takes place by calculating the average price between the two numbers. On the other hand, if the electricity rate price offered by the consumer does not satisfy the amount asked by the prosumer, the transaction does not take place.

As shown in Figure 5, one prosumer can sell energy to various consumers and one consumer can buy energy from more than one prosumer. These transactions were carried out at noon on a fall day. Note that only the consumers who made an energy exchange with prosumers are considered. Moreover, Figure 5 presents the amount of kWh offered by the prosumers and how many kWh they exchanged with the consumers during one hour of the day. In addition, it displays the amount of energy extracted from the grid. For example, building 15 bought solar energy from buildings 19 and 6 simultaneously. In addition, it consumed a total of 31.905 kWh from the grid during that hour. In the case of prosumers 7 and 12, they both sold their energy to consumer 20. Since its demand was satisfied by the energy exchanged with buildings 7 and 12, it did not need any additional energy from the grid. In the third exchange, prosumer 5 sold 10.505 kWh to consumer 10 and, because this amount did not satisfy his needs, he had to take an additional 1.167 kWh from the grid. Nevertheless, in this case study, it has been noted that, at certain times of the day, the total energy generated is bigger than the total energy consumed, which means this additional energy needs to be injected into the grid. Since their demands were satisfied by the energy exchanged with building 20, they did not need any additional energy from the grid.

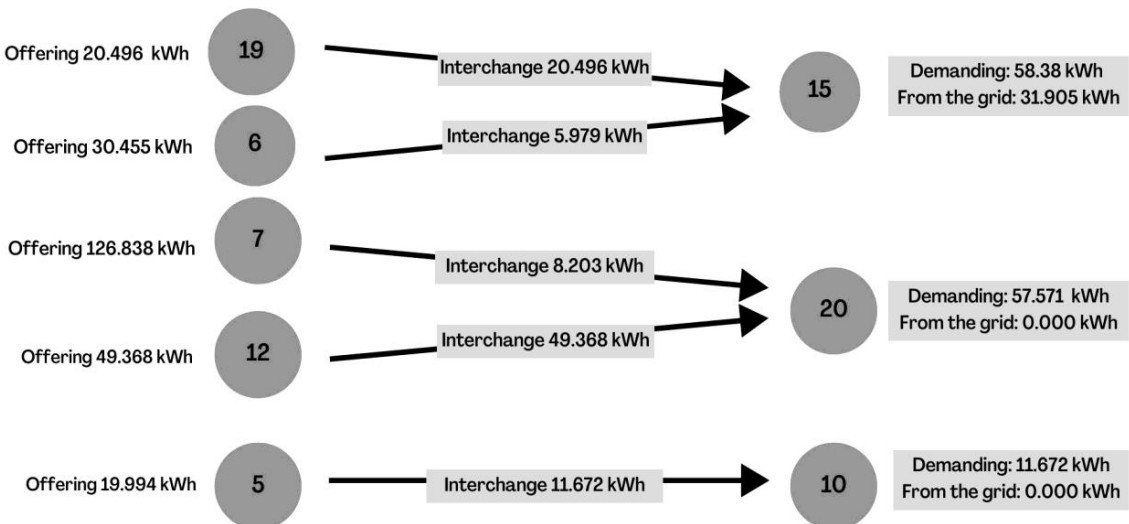

**Figure 5.** Example of the building trading process for an hour in Fall for the P2P model.

## 4. Results

### 4.1. Consumption Profiles

This subsection presents a brief comparison of the total consumption of energy between the different categories of buildings in the four different seasons (Figures 6–9). Given that this study has focused on commercial buildings only, it has been decided to divide them into 14 categories. The categories are the following: Full-Service Restaurant, Hospital, Large Hotel, Large Office, Retail Standalone, Quick Service Restaurant, Secondary School, Warehouse, Small Hotel, Medium Office, Outpatient, Small Office, Primary School, and Retail Strip mall. Throughout the four seasons, the highest consumer category is Retail Strip Malls. Mercantile properties, such as Retail Strip Malls and Enclosed Malls, are composed of multiple connected establishments, which significantly increase the use of electricity compared to other types of commercial buildings. Thus, this category will have one of the highest deficits of energy. Following this category is Large Office with the highest consumption of energy, and afterward comes Warehouse. The category with the least consumption of energy is Small Offices. Overall, the season with the highest consumption was Fall (Figure 9) and the season with the least consumption was Winter (Figure 6). Fall's high energy consumption is likely because during those months the climate is still transitioning from the high temperatures to lower temperatures, and this leads to a more potent cooling system.

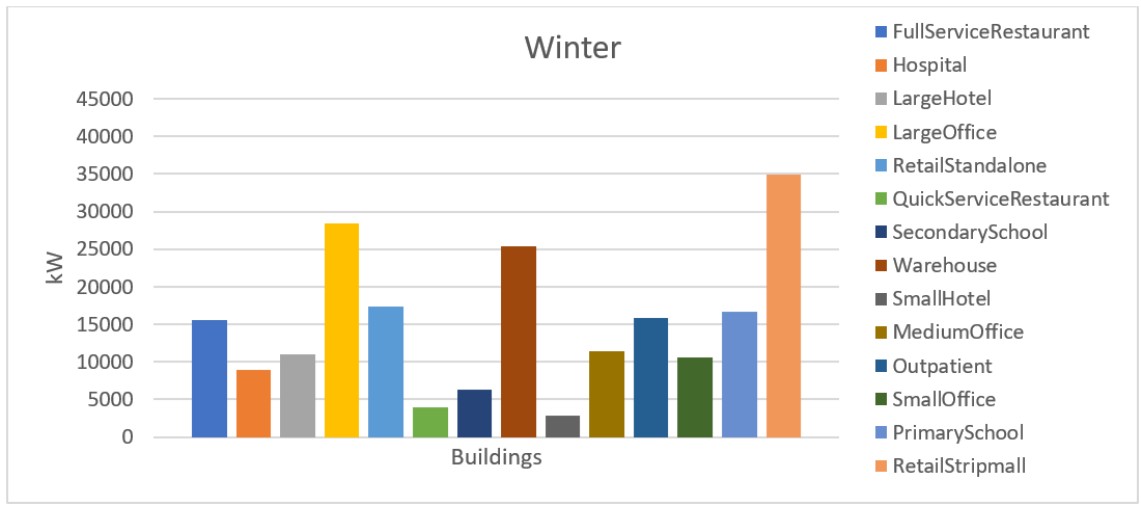

**Figure 6.** Consumption of energy for the different buildings during 7 days of Winter.

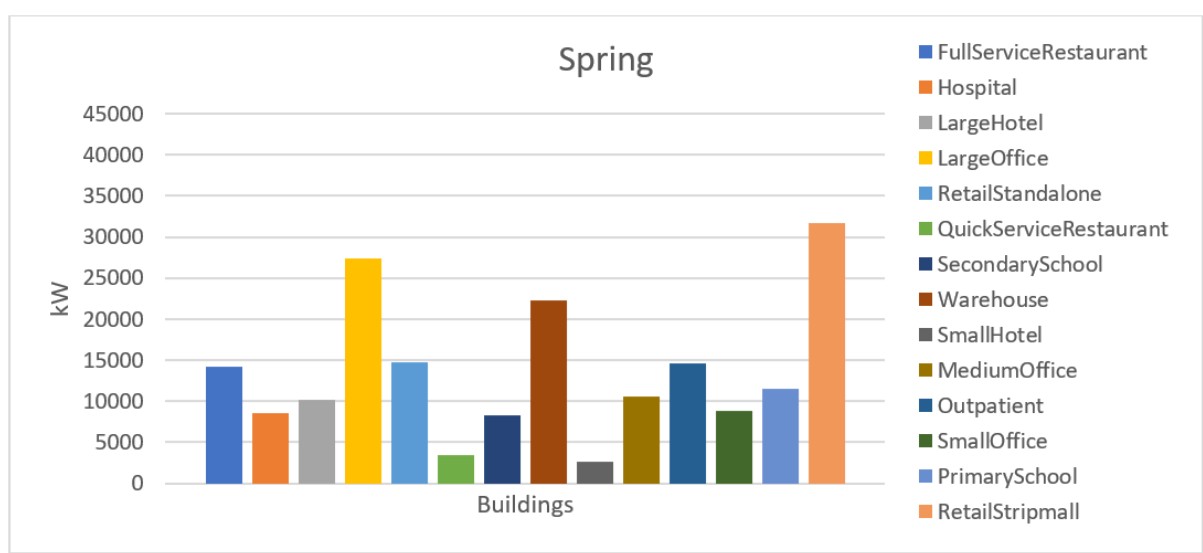

**Figure 7.** Consumption of energy for the different buildings during 7 days of Spring.

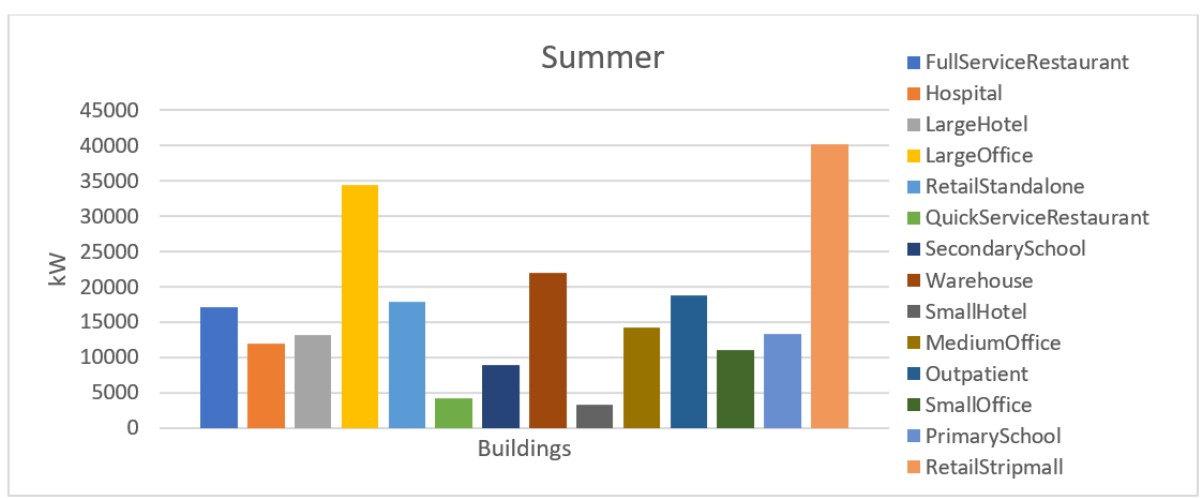

**Figure 8.** Consumption profiles for the different buildings during 7 days of Summer.

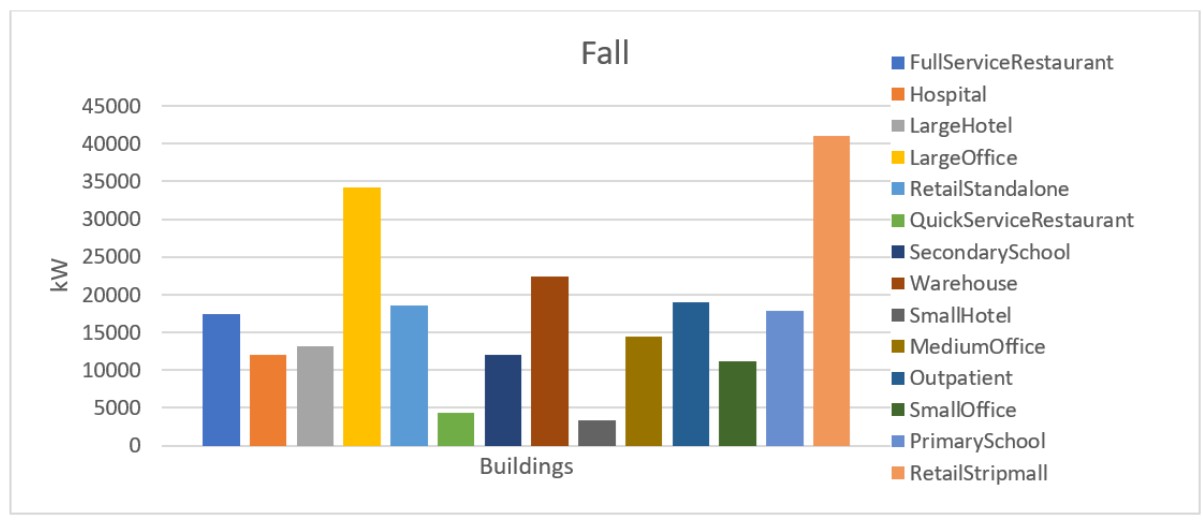

**Figure 9.** Consumption profiles for the different buildings during 7 days of Fall.

*4.2. Consumption and Generation Profiles*

In this subsection, the relationship between the consumption and generation of energy throughout the four different seasons of the year is highlighted. Keeping in mind that only one week was selected from each season, pertaining either to January, April, July, or September, the energy consumption and generation of all 28 buildings are compared in Figure 10. In Figure 10, Winter presented a lower consumption of energy, never surpassing 2000-kWh, compared to those in Summer and Fall. This fact can likely be attributed to the additional effort air conditioners have to make during these months because of the high temperatures outside. During all four seasons, a pattern in peaks of energy generation at midday was identified. This can be accredited to the fact that solar irradiance has a direct relationship with the amount of solar energy generated. In addition, as expected, during the night the solar energy generation is none. The highest consumption at noon on the sixth day studied (Saturday) was 2233.382 kWh during Fall. Following were Summer, Winter, and Spring with 1091.083 kWh, 906.405 kWh, and 902.405 accordingly. The solar generation within the investigated system displays notable seasonal fluctuations. It reaches its pinnacle during the spring season, manifesting an average output of 1571 MWh. Subsequently, the summer period closely trails behind with an average energy yield of 1540 MWh. In contrast, the fall season yields a diminished average generation of 1233 MWh, while the winter season experiences the nadir, producing an average of 1144 MWh. The comparatively subdued photovoltaic yield during the summer months can be attributed to the climatic idiosyncrasies of the Midwest region. Specifically, the summer interval coincides with the region's prevalent rainy season. Consequently, the photovoltaic generation demonstrates a discernible reduction during this period, ostensibly due to increased cloud cover and decreased solar insolation associated with elevated atmospheric moisture levels.

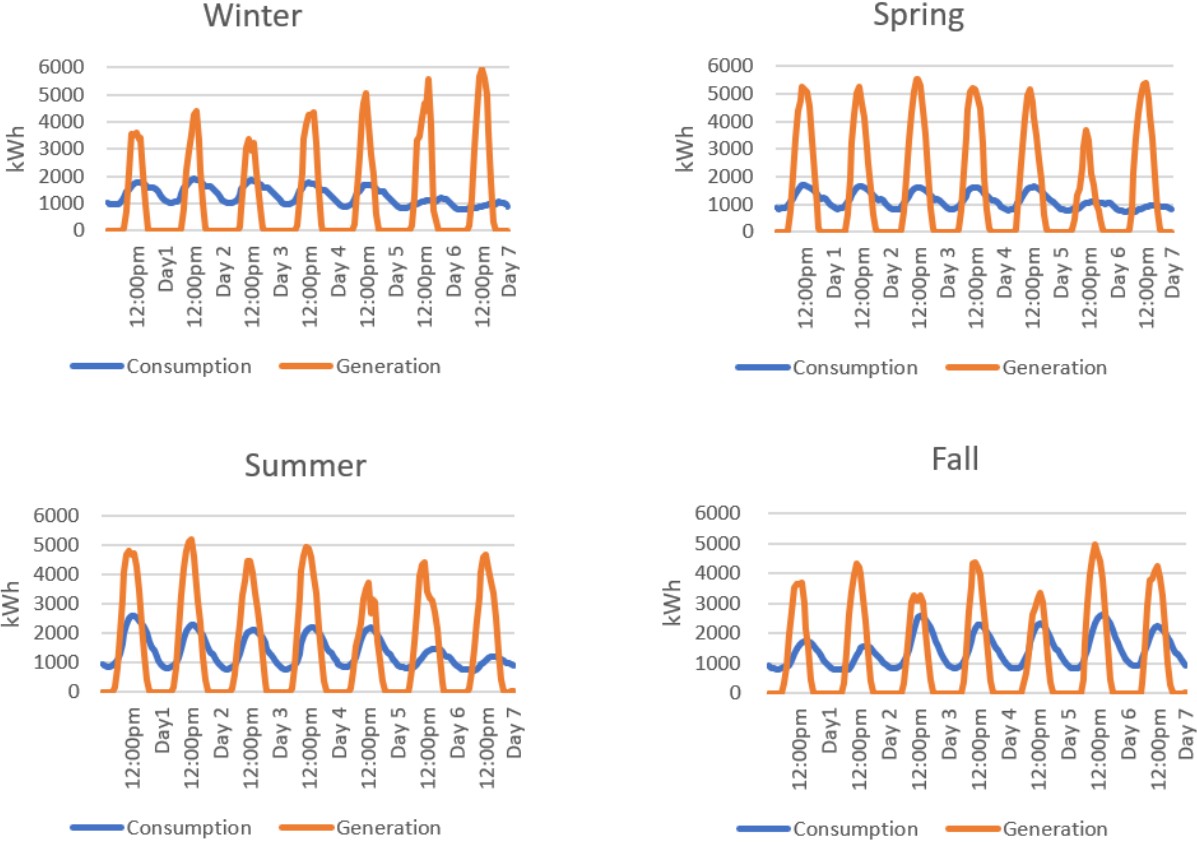

**Figure 10.** Consumption and generation profiles for all seasons.

### 4.3. Supply and Demand of Energy

This subsection presents a comparison between the demand and supply of energy for the four different seasons. Each analysis was made for a whole week starting on Monday and ending on Sunday. It is necessary to clarify that the demand was calculated as the difference between the total consumption and generation of energy. According to Figure 11, Fall and Winter had the highest energy demand. Fall's average peak demand was 1674.9 kWh, while Winter's average peak demand was 1460.733 kWh. Moreover, this can likely be attributed to the need for more potent heat during this time, hence a higher consumption of energy. When analyzing the demand and supply graphs, the trend indicates that, generally, demand peaks are between the hours of 7 pm and 11 pm throughout the four seasons. On the other hand, demand decreased the most during spring. Usually, the peak in energy generation is between 12 pm and 1 pm. Given the generation and consumption peaks do not coincide, a disequilibrium occurs. In more detail, Spring had the steadiest supply of energy and the highest average peak throughout the week, having an average of 3556 kWh at the peak of each day, which was at midday. On the other hand, the lowest supply was during Fall with a peak average of 1504 kWh.

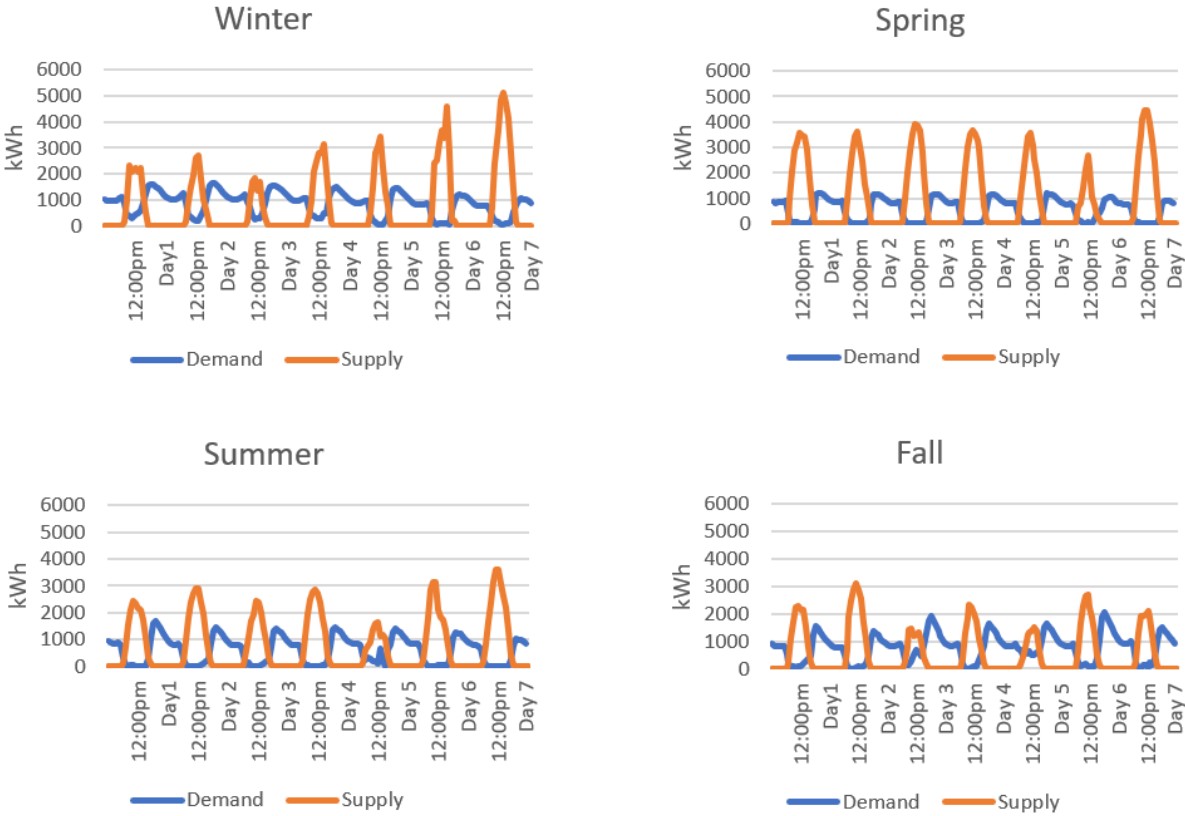

**Figure 11.** Demand and supply for all four seasons.

### 4.4. Technical and Economic Indexes

This subsection displays the findings from examining the four indexes generated for each of the two models and the combination of these. Table 1 presents the results for the indexes for the peaks of power for the four seasons. The indexes for net metering, P2P, and a combination of these have the same values because the peaks of energy in the grid do not change from mechanism to mechanism. Fall had the highest performance of all the four seasons and Spring had the lowest. In other words, Fall was the season that had the least energy peaks in the grid, while Spring was the season with the most frequent energy peaks in the grid. This implies that Fall generated and consumed more or less the same energy, while the deficits and surpluses in energy during the month of Spring were incredibly high

values. Note that values range from 0 to 1 because they have been standardized, where 0 means the lowest performance and 1 means the highest performance.

**Table 1.** The peak of power index for each model and season.

| Peak of Power (kWh) | P2P | Net Metering | P2P & Net-Metering |
|---|---|---|---|
| Winter | 0.0570 | 0.0570 | 0.0570 |
| Spring | 0 | 0 | 0 |
| Summer | 0.2118 | 0.2118 | 0.2118 |
| Fall | 0.3154 | 0.3154 | 0.3154 |

The grid's energy balance is another metric that needs to be examined. Table 2 shows the results for the indexes of energy balance in Fall, Spring, Summer, and Winter. Summer and Spring are the seasons with the best performance when comparing the four seasons in the two mechanisms. For this index, the minimum value was 0, which described the worst performance, and the highest performance would have a value of 1. In the model of P2P, their performance is most similar, differing only by 0.0001. This implies that, during those months, the balance between the energy deficits and surpluses was better. The worst performance for both models and their combination was during Winter. When comparing the net-metering mechanism and the P2P mechanism, it can be observed that P2P had a better overall performance than net metering for the analysis of the energy balance.

**Table 2.** Energy balance index for all seasons and models.

| Energy Balance | P2P | Net Metering | P2P & Net-Metering |
|---|---|---|---|
| Winter | 0.6917 | 0.6435 | 0.6917 |
| Spring | 0.8126 | 0.8014 | 0.8126 |
| Summer | 0.8125 | 0.7974 | 0.8125 |
| Fall | 0.7470 | 0.7130 | 0.7470 |

The economic benefit index was evaluated with the two mechanisms simultaneously and individually. For the combination of net-metering and P2P mechanisms, it was assumed that all the available energy was first offered through the P2P model. After satisfying all the energy demand in the P2P market, the rest was injected into the grid through the net metering program. The index's values range from 0 to 1, where 0 means the worst performance and 1 represents the best performance of the metric. The combination of both mechanisms had the highest performance, with values exceeding 0.99. The second-best economic benefit performance was obtained by the net-metering mechanism. This likely occurs because the economic benefit of the P2P model is limited to the amount of energy demand in the microgrid. Table 3 results show that the season with the best economic performance for P2P and the mechanisms combined was during Spring. On another hand, Fall was the season with the lowest economic index for these two. P2P by itself had the best performance in Winter and the lowest during Spring. The distinction among the performances of these mechanisms might be due to the small scale of the microgrid studied.

**Table 3.** Economic benefit index for all seasons and models.

| Benefit Index | P2P | Net Metering | P2P & Net-Metering |
|---|---|---|---|
| Winter | 0.1322 | 0.9781 | 0.9941 |
| Spring | 0.0000 | 0.9957 | 0.9989 |
| Summer | 0.0203 | 0.9920 | 0.9979 |
| Fall | 0.1181 | 0.9606 | 0.9924 |

The last index evaluated is transaction versus supply and demand. No indexes were calculated for net metering, since there is no supply and demand in their energy transactions. The values in Table 4 range from 0 to 1, where 0 is the least number of transactions and 1 is the highest number of transactions. The highest index for supply and demand did not occur in the same season. Fall has the highest performance in transactions versus the supply index for P2P. This would mean that, during this season, the amount of energy exchanged was more proportional to the amount of energy supplied in the microgrid than in other seasons. However, Winter had the best performance for demand versus supply index. In other words, during Winter, when comparing the energy exchanged in the transactions versus the amount of energy demanded, their values were the closest for all the seasons. In addition, the lowest results did match both supply and demand. Spring had the worst performance in the overall index. This means that during this season the amount of energy exchanged was much higher than the amount of energy demanded or offered in the microgrid. The combination of indexes had the same values as those for P2P, since the inclusion of net-metering does not alter the indexes.

**Table 4.** Transactions vs. supply and demand index for all seasons.

| | P2P | | P2P & Net-Metering | |
|---|---|---|---|---|
| | Transactions vs. Supply | Transactions vs. Demand | Transactions vs. Supply | Transactions vs. Demand |
| Winter | 0.1588 | 0.1353 | 0.1588 | 0.1353 |
| Spring | 0.0307 | 0.0563 | 0.0307 | 0.0563 |
| Summer | 0.0575 | 0.0742 | 0.0575 | 0.0742 |
| Fall | 0.1603 | 0.1184 | 0.1603 | 0.1184 |

Finally, all individual indexes were added to analyze the overall performance of both mechanisms and their combination. In Table 5, it can be observed that the combination of P2P and net-metering model had a better overall performance. Net-metering by itself had the second-best performance, and last came P2P. The season with the best performance for the combination of both mechanisms and P2P is Fall, while the best performance for the net-metering mechanism was during Summer. Due to the high photovoltaic energy generation during this season and considering that all the energy generated is injected into the grid with the net-metering model, one would expect better benefits during this season. In contrast, Spring had the worst performance for the P2P mechanism and the combination of mechanisms. During Spring, P2P had the worst performance in the peak of power index, economic benefit, and transaction vs. demand and supply, which provoked such low values for this mechanism during this specific season. On the other hand, Winter was the season with the lowest overall performance for the net metering mechanism. This situation is because net-metering had a low performance during this season at the peak of power index and the energy balance index.

**Table 5.** Combined indexes for each model and season.

| Combined Indexes | P2P | Net-Metering | P2P & Net-Metering |
|---|---|---|---|
| Winter | 1.1750 | 1.6786 | 2.0369 |
| Spring | 0.8996 | 1.7971 | 1.8985 |
| Summer | 1.1763 | 2.0012 | 2.1539 |
| Fall | 1.4592 | 1.9890 | 2.3335 |

## 5. Discussion

### 5.1. General Discussion

For this case study, a microgrid was created with 28 different building profiles. These buildings' consumption profiles were entirely historical and were altered to include values for each hour. Using random locations throughout the states of Indiana and Illinois, the

generation profiles were created using a PVWatts calculator. These values were generated every hour. One week of each season, starting on Monday and ending on Sunday, were picked to represent the seasonal behavior of the prosumers. Next, a Python simulation was used to model the prosumer bidding process for the exchange of energy. To assess and compare the performance of the P2P model, net metering, and the combination of these, technical and economical indexes were established.

In this study, a comparison of P2P, net metering, and both combined was performed. Most studies have either focused on net metering or P2P, but not many have considered both and the combination of them. For example, Damar's research on P2P for the residential sector [31] proposed a P2P mechanism for a regulated market as an alternative to the current compensation mechanism, net metering. In contrast to our case study, they only considered P2P and net-metering individually and only evaluated the economic benefit the P2P model implies. It was concluded that P2P was more economically suitable than net-metering. In contrast, this study found that net-metering presents a better overall performance than P2P by itself.

The research of Long et al. on P2P energy sharing in a microgrid [32] introduced a two-stage control method to implement community energy sharing through a P2P model. Unlike this paper's focus, Long and his colleagues did not take into consideration P2P as a complement to current net-metering policies. This could be attributed to the phasing out of net-metering policies worldwide. Part of this study's focus was on the impact the seasonal component has on energy consumption and generation.

The study of An et al. [33] discussed P2P as a profitable energy trading strategy and concluded that the month with the most active energy trading was May. Our results suggested that the season with the highest amount of transactions was Winter (January). It should be noted that the study of An et al. was simulated in South Korea and in the residential sector, while ours was based on the states of Indiana and Illinois and in the commercial sector. Moreover, electricity rates vary by on and off-peak hours in South Korea, while this policy is usually optional in the States.

Soto et al. [2] performed a similar case study where P2P in a residential sector, P2P EV, and net-metering were compared based on their economic and technical performances. Like this study, their findings indicated that the season with the highest amount of transactions was Winter (January).

On the other hand, Hutty et al. [34] found that the season's best performance in improving the self-sufficiency ratio and energy bill savings was Fall. Hutty's paper demonstrated how P2P and electrical vehicles can provide a better energy balance and save on energy bills in a simulation of a 50-house microgrid in London. For this study, it was found that the second season leading to energy balance was Summer, after Spring. The divergence in the results could be attributed to the difference in geographical location and it should be noted that Hutty did not consider Spring in the comparison among the seasons.

Similarly, Wang and his colleagues [7] found that a P2P model implemented in a microgrid creates a better energy balance in the power grid. In addition, this study did not find a significant difference in energy peaks when introducing P2P to the energy market, and neither did other studies [31–33]. The low performance of P2P on the energy peak index could be attributed to the fact that readings were done hourly. Future research could work with smaller intervals and observe if energy peaks decrease and increase within one hour.

Finally, when comparing this research with other studies it is found that P2P by itself has a lower performance than net metering. In contrast, Soto et al. found that P2P had a higher overall performance than net-metering, although if both were combined they would achieve the highest performance according to the economic and technical indexes.

Overall, this study has found some divergent and similar results, considering the different indexes used, with other studies on the performance of the mechanism P2P. In addition, this research considered the impact of the seasons on the performance of both mechanisms since it was simulated for states of the Midwest of the United States, which

have very variable weather throughout the seasons. The drastic changes in the weather directly impact the generation and consumption of energy during acute seasons, such as summer and winter.

*5.2. Limitations and Future Research*

Various limitations are associated with this case study and its development. Initially, this case study was entirely performed utilizing a simulation. Additionally, it considers the ideal case where all of the users in the microgrid own PV systems. Moreover, this study only takes into account a limited number of commercial buildings which participate in this microgrid. An additional limitation is the number of days considered in this study which is restricted to only seven days of each of the four seasons, limiting the appreciation of consumption and generation behaviors. Furthermore, it does not consider either residential or industrial prosumers for the bidding interactions.

P2P models have a significant potential to reframe the energy market in the United States. This study only focused on a limited number of commercial buildings in a limited geographical region. Therefore, there are a considerable number of areas where further research can be developed. One of the limitations of this case study is the time frame selected to evaluate the economic and technical indexes. In future studies, a wider time frame can be selected to observe in more detail the behavior of these mechanisms. In addition, this study can evaluate consumption profiles, generation profiles, and energy trading in smaller intervals of time. For example, instead of performing the biddings simulation every hour, one could perform it every 15 min. Shortening the intervals might allow for the peaks of energy to be studied more in-depth and detect if P2P influences these energy peaks in shorter intervals of time. Moreover, the geographical area selected for the development of this study can be expanded to compare the weather of other regions to that of the Midwest of the United States. Another option for future research is the expansion of the customer segment, hence one could integrate residential prosumers with commercial prosumers. Likewise, the complexity of the grid can be increased with the addition of energy storage, such as batteries. In light of the contemporary landscape, in which there are algorithms capable of solving simulation dilemmas in a matter of seconds or even less, it becomes imperative to integrate such algorithmic paradigms by introducing novel and more complex models in future research.

In the context of future research on P2P energy trading systems, the work of Shukhobodskiy and colleagues offers a promising foundation for integrating advanced control methodologies into P2P trading networks [35]. Shukhobodskiy et al. [35] introduce an innovative adaptive control strategy for hybrid energy storage systems encompassing various components, such as thermal storage reservoirs, heat pumps, storage heaters, a photovoltaic array, and a battery. Notably, the proposed algorithm based on the RED WoLF project [36] demonstrates substantial carbon reduction improvements compared with the previous version of the algorithm, showcasing potential avenues for enhanced emissions reduction within P2P systems. The manuscript's insights can significantly inform forthcoming research seeking to optimize energy utilization and emissions reduction in P2P energy trading systems by leveraging hybrid energy storage technologies and advanced control strategies.

Additionally, for future research, datasets such as "electricity maps" [37] along with "carbon intensity" [38] collectively serve as invaluable repositories for forward-looking research delving into the convergence of P2P energy trading and the fundamental metrics of carbon intensity, including electricity price. Through real-time and historical data sets, these platforms provide a complex picture of global sources of electricity generation and associated carbon emissions, establishing a fundamental framework for characterizing the carbon footprint of conventional power generation. This framework sets the stage for comparative analyzes of P2P business scenarios, allowing for an assessment of their environmental impact and clarifying strategies for refining business dynamics to reduce carbon emissions. Leveraging visualizations and empirical data, these platforms power

impact analysis, policymaking, and simulation modeling, coming together to expose the promise of P2P energy trading as an environmentally savvy mechanism for orchestrating sustainable energy transitions.

## 6. Conclusions

One of the study's main conclusions is the ineffectiveness of the peak of energy implementing a P2P mechanism into a microgrid. When P2P was compared to net-metering and their combination, there was no difference in the peak of the energy index for all seasons. This could be because peak generation and peak consumption hours continue to differ. In future studies, energy storage implementation could help reduce the grid's heavy load and energy peaks.

Second, the implementation of a P2P model improved the overall energy balance of the grid. When the sensitivity of the models to the seasons was considered, it was discovered that Spring had the best performance for energy balance. Summer followed with an almost exact performance index, indicating that generation profiles during these months coincide significantly with commercial building consumption profiles. The use of a P2P model helps increase the grid's energy balance due to the exchange of energy among buildings, which puts surplus energy to use at different times and reduces the load on the grid. It should be noted that the energy peak index performed the worst in the spring, but when energy trading was implemented throughout P2P, the balance of energy on the grid achieved the highest balance during this season.

Third, it was found that a combination of P2P and net-metering provided the highest economic benefit for prosumers. The implementation of these two mechanisms signifies a meaningful economic advantage for prosumers who wish to sell their excess energy. This behavior was observed in all seasons, having the highest performance during Spring. Adopting these models implies better energy market prices because it permits the introduction of more competitive prices and an accelerated return on investment for prosumers.

Fourth, transaction vs. supply and demand indexes did not present significant differences among them. Both supply and demand had lower indexes during Spring and Summer and higher indexes during Winter and Fall. The lower indexes for Summer and Spring might indicate there is either more demand than energy exchanges or more supply than energy exchanged. Hence, the P2P mechanism can be optimized to better manage demands and supplies of energy in the grid during these seasons.

Finally, the combination of these indexes indicates that P2P combined with net-metering had the best overall performance. This proves that P2P is an effective alternative or complement to traditional net-metering programs when considering economic and technical indexes. This study will encourage policymakers to create better policies to incentivize and support innovative mechanisms such as P2P that support the generation of renewable energies.

Peer-to-peer generation exhibits diverse applications beyond microgrid contexts. In residential communities, it enables the direct sharing of surplus energy between home-owners equipped with solar panels, fostering collaboration and efficient energy utilization. P2P can also revolutionize electric vehicle (EV) charging by allowing EV owners to trade electricity from their vehicle batteries, optimizing charging dynamics and grid balance. Renewable cooperatives can leverage P2P to form collective investments in renewable installations, sharing generated energy among members. Industries can enhance energy management within complexes by redistributing surplus energy to facilities with varying demands. Moreover, P2P suits remote areas, enabling localized energy-sharing networks using renewables, and addressing energy access challenges. In commercial settings, energy sharing between buildings can optimize resource utilization. This aligns with smart city initiatives, transforming urban energy distribution. Energy trading platforms driven by P2P can introduce competitive pricing and consumer empowerment. P2P systems are adaptable for temporary events and disaster recovery, providing prompt energy access. Agriculture can benefit from energy sharing for irrigation and processing. Multifamily

housing and research institutions can improve energy distribution efficiency. These applications underscore P2P's flexibility in enhancing energy distribution, resilience, and sustainability across sectors.

**Author Contributions:** Conceptualization, E.A.S.; methodology, E.A.S.; validation, E.A.S.; formal analysis, A.V.O. and A.H.; writing—original draft preparation, A.V.O. and A.H.; writing—review and editing, E.A.S. and L.B.; supervision, L.B.; project administration, E.A.S.; funding acquisition, L.B. All authors have read and agreed to the published version of the manuscript.

**Funding:** National Science Foundation Award #2050451.

**Data Availability Statement:** Data sharing not applicable.

**Conflicts of Interest:** The authors declare no conflict of interest.

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
