# Peer review of "Commercial Level Analysis of P2P vs. Net-Metering Comparing Economic and Technical Indexes"

_2673-4117, doi:10.3390/eng4030129_

Round 1

Reviewer 1 Report

The paper is about comparing peer-to-peer and net-metering and combining both.

The central tool seems to be PyMarket. However, this tool seems quite simple. There are many P2P trading mechanisms. How can the comparison be complete when PyMarket only includes few of these mechanisms?

I do not consider using an existing tool and defining a few metrics as high novelty. The authors have to justify clearly why this paper should be published.
Please explain clearly what the contribution of this paper is aside from using an existing tool, defining a few metrics and producing results.

More specific comments:
- Section 1.2 does not discuss the research gap and the novelty of the paper. This does not happen before Section 5. I suggest, to move most of Section 5 to Section 1.2. A discussion Section should be about the results, which may include a sensitivity analysis or limitations of the methodology. The research gap has to be discussed closer to the beginning of the paper. The novelty has to be clearly stated
- Figure 1 is blurred.
- In Figure 2, the text is blurred. The figure is not labelled and it does not have a caption.
- Figure 2 (should be Figure 3) on page 11: Add the information (e.g. in the caption) that this is about buildings. Only numbers in circles are shown.
- Figure 4 on page 12 is offset to the right of the page. The title says "Spring", but the caption talks about both spring and fall (autumn).
- Figure 3 follows after Figure 4 on page 12.
- Figure 4 on page 13 is also about spring. Figure 6 on page 13 is about fall. Hence, it seems that Figure 4 on page 12 is obsolete. Please check.
- The text in Section 4 is often quite speculative ("could be", "could mean"). Please use clear statements with proper reasoning.
- Page 14: It is hard to believe that the output of PV is highest in winter. Please check this again properly. Maybe the chosen seven "representative days" were not that representative after all.
- Table 1: Why is the peak of power index always 0 for spring?
- The conclusion is too long. Part of it should be in the discussion section (e.g. limitations).

Finally, the presentation of the paper is unacceptable.
The order of figure labels is wrong. Some figures are not labelled, while the label "Figure 4" appears twice. Some figure captions are missing, some are wrong.
In the text, the error "Error! Reference source not found." appears quite a few times. This is not supposed to happen.

The wording is sometimes strange. The language is not very good. The paper also contains typographical errors (punctuation, spaces, etc.).
A moderate English check is strongly recommended.

Reviewer 2 Report

1 Lines 103-104 ( Nonetheless, smart devices are not without their drawbacks. Since they usually provide real-time data- they have a limited capacity for power forecast. )  Is it power generation or power consumption forecast? For example the RED WoLF project (https://vb.nweurope.eu/projects/project-search/red-wolf-rethink-electricity-distribution-without-load-following/) not only allows power forecast of power consumption or generation but also utilises the MPARC controller that allows to operate in real time  in order to schedule storage of energy with the aid of hybrid electrochemcial and thermal storage reservoirs, the could be seen in listed scientific publications as well as technical reports and newsletters.The power could be send back to the grid or shared between facilities as well as applied to commercial and public facilities e.g. Shukhobodskiy et al. 2023 https://doi.org/10.1016/j.egyai.2023.100287 with demand management layer could be added on top. Please discuss the aspect on real time controlled smart energy systems that allow both P2P and net-metering, in more details.

2. Line 450 some references are missing please correct it, please also correct that throughout the text.

3. Line 734 the are algorithms the allow simulation to be performed on sub-second basis even less than 1 ms 

4. Please improve conclusion by adding more applications of P2P energy genercation.

5. websites lie carbon intensity uk https://carbonintensity.org.uk/ allows the grid carbon intensity information on intrahour basis both forecast and actual, whereas electricitymap https://app.electricitymaps.com/ allows not only carbon intensity information but also electricity consumption and its price. Please incorporate discussion with regards to this services.

Reviewer 3 Report

This manuscript proposes an analytical comparison between the use of the peer-to-peer mechanism, the net-metering mechanism, and a combination of these in the commercial sector. The authors need to address the following comments:

a. The presentation of the manuscript needs to be done in a clearer way. The authors should be more straightforward by condensing the text.

b. The objective should be clearly described in the abstract section.

c. The authors need to define abbreviations before using them (line 12, 116 – PV, P2P acronyms are used without being defined before).

d. The authors should comment on data presented in figures and tables.

e. Figure that shows the consumption of energy for the different buildings during 7 days of Spring appears twice (page 12 and 13).

f. Some references are not properly defined (line 450, 454, 486, 507, 530, 547, …).

g. The conclusions section should be improved.

h. Impersonal writing should be used consistently throughout the manuscript.

The use of English needs to be checked and manuscript revised (Line 118, …).

Round 2

Reviewer 1 Report

Thank for incorporating my comments and suggestions.
The quality of the paper is much higher now.

Reviewer 2 Report

Dear Authors,

Thank you for providing correction, however there are a few things to further correct.

1 Unfortunately I was unable to find section with discussion as were highlighted in the reply "Response: Thank you for the feedback. This topic was incorporated in the introduction section and the study of the RED WoLF project was added as a citation." Please correct it.

2.  Line 734 the are algorithms the allow simulation to be performed on sub-second basis even less than 1 ms Response: Thank you for the feedback. Information about the speed of the algorithms in the conclusions section was added.  Unfortunately I was unable to find clear statement, that there are methods allowing prompt operation in the conclusion. Please correct it 

Author Response

Thank you for providing correction, however there are a few things to further correct.

  1. Unfortunately I was unable to find section with discussion as were highlighted in the reply "Response: Thank you for the feedback. This topic was incorporated in the introduction section and the study of the RED WoLF project was added as a " Please correct it. Response: Thank you for the feedback. The RED WoLF project was added in section 5.2

“In the context of future research on P2P energy trading systems, Shukhobodskiy and colleagues' work offers a promising foundation for integrating advanced control methodologies into P2P trading networks [35]. Shukhobodskiy et al. [35] manuscript introduces an innovative adaptive control strategy for hybrid energy storage systems encompassing various components like thermal storage reservoirs, heat pumps, storage heaters, a photovoltaic array, and a battery. Notably, the proposed algorithm based on the RED WoLF project [36] demonstrates substantial carbon reduction improvements compared with the previous version of the algorithm, showcasing potential avenues for enhanced emissions reduction within P2P systems”

  1. Line 734 the are algorithms the allow simulation to be performed on sub-second basis even less than 1 ms Response: Thank you for the feedback. Information about the speed of the algorithms in the conclusions section was added. Unfortunately I was unable to find clear statement, that there are methods allowing prompt operation in the conclusion. Please correct

Response: Thank you for the feedback. The topic was included in section 5.2

“In light of the contemporary landscape in which there are algorithms capable of solving simulation dilemmas in a matter of seconds or even less, it becomes imperative to integrate such algorithmic paradigms by introducing novel and more complex models in future research.”

Reviewer 3 Report

The revised paper may be considered for publication.

Round 3

Reviewer 2 Report

The manuscript is improved. I would like to thank authors for corrections and recommend it to publication.